# Multilineage Dysplasia as Assessed by Immunophenotype in Acute Myeloid Leukemia: A Prognostic Tool in a Genetically Undefined Category

**DOI:** 10.3390/cancers12113196

**Published:** 2020-10-30

**Authors:** Francesco Mannelli, Sara Bencini, Matteo Piccini, Giacomo Gianfaldoni, Maria Ida Bonetti, Benedetta Peruzzi, Roberto Caporale, Barbara Scappini, Fabiana Pancani, Vanessa Ponziani, Leonardo Signori, Michela Zizza, Francesco Annunziato, Alberto Bosi

**Affiliations:** 1Struttura Operativa Dipartimentale Ematologia, Azienda Ospedaliero-Universitaria Careggi, 50134 Firenze, Italy; sarabencini1179@gmail.com (S.B.); piccinim@aou-careggi.toscana.it (M.P.); gianfaldonig@aou-careggi.toscana.it (G.G.); marida.bonetti@gmail.com (M.I.B.); scappinib@aou-careggi.toscana.it (B.S.); fabianapancani@virgilio.it (F.P.); vanessa.ponziani@uslnordovest.toscana.it (V.P.); leonardo.signori@unifi.it (L.S.); michela.zizza@unifi.it (M.Z.); alberto.bosi@unifi.it (A.B.); 2Centro Ricerca e Innovazione Malattie Mieloproliferative (CRIMM), AOU Careggi, 50134 Firenze, Italy; 3Centro Diagnostico di Citofluorimetria e Immunoterapia, AOU Careggi, 50134 Firenze, Italy; peruzzib@aou-careggi.toscana.it (B.P.); caporaler@aou-careggi.toscana.it (R.C.); francesco.annunziato@unifi.it (F.A.)

**Keywords:** acute myeloid leukemia, immunophenotype, multilineage dysplasia, AML with myelodysplasia-related changes

## Abstract

**Simple Summary:**

The prognostic role of multi-lineage dysplasia is still debated in acute myeloid leukemia. The aim of our work was to study dysplasia by a technique alternative to the conventional morphological method, which is multi-parameter flow cytometry. To this end, we used an immune-phenotypic score (IPS), able to estimate dysplasia by the extent of deviation from normal profile, obtained in a control group. IPS provided no insight into prognosis when considered overall nor within well-defined genetic categories. Of interest, IPS-related dysplasia conveyed significant prognostic information when we focused on genetically undefined patients, triple-negative for NPM1, FLT3 and CEBPA. This category still represents a non-negligible fraction of patients, that lack specific molecular features either for targeted drugs or for proper risk assessment. In this context, our data could help address the relative unmet needs in treatment strategy, and provide insight into response prediction in the rapidly evolving therapeutic scenario of AML.

**Abstract:**

Acute myeloid leukemia (AML) “with myelodysplasia-related changes (MRC)” is considered a separate entity by the World Health Organization (WHO) classification of myeloid neoplasms. While anamnestic and cytogenetic criteria provide objective attribution to this subset, with clear unfavorable prognostic significance, the actual role of multi-lineage dysplasia (MLD) as assessed by morphology is debated. The aim of our work was to study MLD by a technique alternative to morphology, which is multiparameter flow cytometry (MFC), in a large series of 302 AML patients intensively treated at our Center. The correlation with morphology we observed in the unselected analysis reiterated the capability of the MFC-based approach at highlighting dysplasia. MLD data, estimated through an immune-phenotypic score (IPS), provided no insight into prognosis when considered overall nor within well-defined genetic categories. Of interest, IPS-related dysplasia conveyed significant prognostic information when we focused on genetically undefined patients, triple-negative for *NPM1*, *FLT3* and *CEBPA* (TN-AML). In this context, the lack of dysplastic features (IPS_0) correlated with a significantly higher CR rate and longer survival compared to patients showing dysplasia in one or both (neutrophil and erythroid) cell lineages. The impact of IPS category maintained its validity after censoring at allogeneic HSCT and in a multivariate analysis including baseline and treatment-related covariates. In a subgroup featured by the lack of genetic determinants, our data could help address the relative unmet needs in terms of risk assessment and treatment strategy, and provide insight into prediction of response in the rapidly evolving therapeutic scenario of AML.

## 1. Introduction

Acute myeloid leukemia (AML) “with myelodysplasia-related changes (MRC)” is considered a separate entity by the World Health Organization (WHO) classification of myeloid neoplasms [1,2]. The attribution to this category would point out the clonal evolution of AML from a previous myeloid disorder, a process that generally indicates an unfavorable prognosis. This can rely on three different criteria: anamnestic (previous diagnosis of myelodysplastic syndrome (MDS) or myeloproliferative neoplasm); cytogenetic (presence of specific karyotype abnormalities); morphological (evidence of multilineage dysplasia (MLD)). Anamnestic and cytogenetic criteria are not usually subject to controversy because of the objectivity of their evaluation and of the clear demonstration of the inherent unfavorable prognosis, which is, in turn, preparatory for an appropriate therapeutic approach [3,4,5]. Major issues regard AML with MLD, both for its correct classification and its actual prognostic significance. The available literature yields conflicting results, possibly for technical and biological reasons [3,4,5,6,7]. The morphological assessment of residual hematopoiesis at AML diagnosis is primarily flawed by operator-dependency. Its specific application is even more complicated in this setting because residual non-blast cells can be very few at diagnosis. Biologically, MLD might convey insight into a pre-existing clonal hemopoiesis but it could also merely result from pathologic differentiation/maturation by the leukemic clone. Further, the expansion of knowledge about genetic background of AML calls for a reconsideration of MLD role. In well-defined genetic entities, such as AML-bearing *NPM1* or *CEBPA* (bi-allelic) mutations, the role of MLD has been downsized and the therapeutic strategies should not base upon it in this context [8,9,10,11,12]. In less characterized AML subsets, it is still debated if MLD prognostic significance is meaningful enough to support important clinical decisions as the allocation to allogeneic stem cell transplantation (HSCT) [13]. The aim of our work was to study MLD by a technique alternative to morphology, which is multiparameter flow cytometry (MFC), progressively emerging as a useful method to assess dysplasia, particularly in the MDS setting [14,15,16,17]. Our rationale was that an MFC-based evaluation could get further insight into MLD’s actual significance, especially in distinct genetic subsets.

## 2. Patients and Methods

### 2.1. Patients

Patients entering the study had a diagnosis of untreated non-promyelocytic AML, based on morphological, immunophenotypic, and molecular criteria, and were intensively treated as specified in the Appendix A. The study was approved by the local institutional review board (protocol number: 2013/0034874). Written informed consent was obtained from trial patients in accordance with the Declaration of Helsinki. Enrolment criteria required (1) intensive treatment, (2) availability of immunophenotypic data at diagnosis, and (3) signed informed consent.

### 2.2. Morphology

Morphological revision of AML cases was carried out on bone marrow smears, stained with the May–Grunwald–Giemsa method. Dysplasia was assessed according to WHO [1,2]. The definition of multi-lineage dysplasia (MLD) was based on the presence of dysplasia (>50% of dysplastic cells per lineage) in at least two cell lineages in bone marrow smears. Dysgranulopoiesis was defined as ≥50% of ≥10 polymorphonuclear neutrophils being agranular or hypogranular, or with hyposegmented nuclei (pseudo Pelger-Huet anomaly). At least 25 cells were evaluated, but usually 100 cells were counted. Dyserythropoiesis was defined as ≥50% of dysplastic features in ≥25 erythroid precursors: megaloblastoid aspects, karyorrhexis, nuclear particles, or multinuclearity. Dysmegakaryopoiesis was diagnosed when ≥3 megakaryocytes or ≥50% in ≥6 cells showed dysplastic features such as micromegakaryocytes, multiple separated nuclei, or very large single nuclei.

### 2.3. Karyotype

Cytogenetic analysis was performed on BM cells at diagnosis according to the International System for human Cytogenetic Nomenclature [18]. Cytogenetic risk was defined according to Medical Research Council (MRC) [19].

### 2.4. Molecular Genetics

The presence of the *NPM1* mutation was defined by immunohistochemical criteria, that is demonstration of aberrant cytoplasmic expression of *NPM1* [20] or by mutational analysis for *NPM1* gene mutations [21]. The presence of *FLT3* internal tandem duplications (ITD) and Tyrosine Kinase Domain (TKD), and *CEBPA* mutations were investigated as described [21,22,23].

### 2.5. Flow Cytometry

In order to assess MLD by immunophenotype, we adapted to AML an approach previously described for MDS [14]: dysplasia was appraised for neutrophil and erythroid compartments through an immuno-phenotypic score (IPS) including 17 parameters (13 for neutrophil and four for erythroid compartment). Cell compartments were considered not assessable for dysplasia when not detectable as at least 0.01% of total BM cells. For data analysis, Infinicyt (Cytognos SL, Salamanca, Spain) software was used. Technical details were published previously [11] and summarized in the Appendix A. Briefly, neutrophil and erythroid cell compartments were identified on the basis of forward (FSC) and sideward (SSC) light scatter characteristics and their reactivity for CD45. Parameters included by IPS were expressed as percentage of positive cells for an antigen within a cell compartment and/or its mean fluorescence intensity (MFI; arbitrary relative linear units, scaled from 0 to 10^4^, normalized upon a control). A score was attributed to each parameter included in the IPS depending on the extent of deviation from normal phenotypic profile, as defined in parallel in control groups. Single cell lineages of AML cases were considered to be dysplastic when their relative IPS value was higher than mean IPS + 2 standard deviations (SD) in controls.

### 2.6. Definitions

Complete remission (CR), non-responsive disease (NR), early induction death (ED), disease-free survival (DFS), and overall survival (OS) were defined according to standard criteria [24]. For risk definition, patients were stratified according to the European Leukemia Net (ELN) 2010 system in post hoc analysis [25].

### 2.7. Statistical Aspects

Pairwise comparisons between patient characteristics were performed using the Mann–Whitney test or the Kruskal–Wallis test for continuous variables and Pearson’s chi-squared test or Fisher’s exact test for categorical variables. Survival was estimated with the Kaplan–Meier method and long-term outcomes were compared with the log-rank test. The Cox proportional-hazards model was applied to estimate hazard ratios with 95% confidence intervals (CI) for disease-free survival (DFS), i.e., the interval from CR to relapse or death, overall survival (OS), i.e., the interval from study entry to death, and event-free survival (EFS), i.e., the interval from study entry to primary refractory disease, relapse or death, whichever occurs first) in both univariate and multivariate contexts. In order to rule out an impact by allogeneic SCT, we censored patients receiving allogeneic SCT at the date of transplant in a further analysis. All *P* values were two-sided, and a 5% significance level was set. Statistical analyses were performed using R version 3.5.0 and SPPS version 26.

## 3. Results

### 3.1. Characteristics of Patients and Treatment Flow

From April 2004 to April 2017, 326 patients affected by AML met the inclusion criteria. Maturing cell compartments were both detectable (i.e., ≥0.01% of global cells) and evaluable for dysplasia in 302 (92.6%) patients. Their clinical and biological characteristics are detailed in Table 1. As per induction treatment, 209 (69.2%) and 93 (30.8%) received standard-dose (SDAC) and high-dose cytarabine (HDAC) containing regimen, respectively. The comparison between induction treatment groups did not show any relevant unbalances for main clinical and biological features (Appendix A) nor differences in survival estimates (Appendix A). Overall, at the evaluation of responses after first induction course, 182 patients achieved CR (60.2%), 15 experienced an ED (5.0%), and 105 were declared NR (34.8%). When evaluated after a second chemotherapy course administered to NR patients, the CR rate increased to 76.5% (*n* = 231/302). One-hundred-and-twenty patients received an allogeneic transplant, of whom 88 were in first CR as part of the consolidation phase.

### 3.2. Study of MLD by Immuno-Phenotype

The median IPS in the overall cohort was 3.5 (range 0.0–13.5), as the result of the sum of phenotypic score in neutrophil (median 2.0 (0.0–10.5)) and erythroid (1.0 (0.0–5.0)) lineage in individual cases. We explored the distribution of IPS values according to morphological findings: a parallel trend was observed between IPS values and the number of dysplastic cell lineages as stratified by morphology (Figure 1A). Median phenotypic scores in single lineage (neutrophil or erythroid) were higher when the same lineage was judged as dysplastic at morphology (Appendix A); the trend was not significant for erythroid cells, possibly because of a narrower range of variability. When investigated across different AML sub-classifications, non-significant trends toward higher median IPS values were observed for adverse karyotype, *CEBPA*-DM and WHO-defined MRC (Figure 1B,C and Appendix A). Within the latter group, the distinction between MDS-related cytogenetic abnormalities versus morphologically-assessed MLD did not show any relevant difference in IPS (Figure 1D). A limited sample size (*n* = 4) must be acknowledged for the range of values in tAML. Based on normal ranges defined in the control group, 74 (24.5%) patients showed both cell lineages as phenotypically normal (IPS_0), whereas 133 (44.0%) and 95 (31.5%) had one (IPS_1) and two (IPS_2) dysplastic lineages, respectively. No significant clinical–biological difference emerged from the comparison of these three groups, including their distribution in selected WHO- and genetically defined categories, except for a lower platelet count in IPS_1 and IPS-2 categories (Appendix A).

### 3.3. Correlation with Outcome in the Overall Cohort and within Selected Subgroups

The single-course CR rate was 64.9% (*n* = 48/74), 54.9% (*n* = 73/133) and 64.2% (*n* = 61/95) in IPS_0, IPS_1 and IPS_2, respectively (*p* = 0.956). After a second cycle, CR rate increased up to 83.8% (*n* = 62/74), 70.7% (*n* = 94/133) and 78.9% (*n* = 75/95) in IPS_0, IPS_1 and IPS_2, respectively (*p* = 0.576). We then assessed whether IPS results affected the risk of relapse and survival estimates. No significant effect by IPS was observed on outcome as assessed by DFS, OS or EFS in the overall cohort (Figure 2). In the analysis of OS, a non-significant trend toward better outcome was observed for IPS_0 patients (in single comparisons, *p* = 0.083 and *p* = 0.024 versus IPS_1 and IPS_2, respectively). We searched for any interaction of IPS and prognosis within WHO- and genetically defined disease subsets. IPS did not show a significant prognostic impact in AML with recurrent genetic abnormalities (RGA) nor in AML–MRC (Appendix A). Within non-otherwise-specified (NOS) AML, a trend for longer survival emerged in IPS_0 patients, reaching statistical significance for EFS (Appendix A). IPS did not correlate with survival in cytogenetic categories (favorable, intermediate, adverse), nor in *NPM1*-mutated and *FLT3*-ITD subgroups (Appendix A).

### 3.4. Correlation with Outcome in the Triple-Negative Subset

An impact on prognosis emerged when we focused on genetically undefined patients, that means with intermediate-risk or lack of growth karyotype, and triple-negative for *NPM1*, *FLT3*-ITD and *CEBPA* (TN-AML). The survival curves showed a parallel trend between IPS and prognosis: the higher was the number of dysplastic lineages, the shorter was the survival as assessed by DFS, OS, EFS (Figure 3A–C). Since IPS_1 and IPS_2 had very similar survival, these patients were gathered and compared to IPS_0 subgroup in the following analyses. As per baseline characteristics, IPS_0 showed a significantly lower WBC count and incidence of normal karyotype compared to IPS_1–2 (Table 2). Overall CR rate was 90.9% (*n* = 20/22) and 68.9% (*n* = 42/61) in IPS_0 and IPS_1-2, respectively (*p* = 0.048, OR = 4.52; 95% CI 0.96–21.3). The IPS group affected prognosis, as demonstrated by DFS, OS and EFS (Figure 3D–F). Analogous results were found after censoring at allogeneic HSCT (Appendix A). Subgroup analysis within TN-AML category confirmed the validity of this result across different clinically and genetically defined subsets (Figure 4 and Appendix A). Of interest, the presence of multi-lineage dysplasia, as assessed by morphology, did not influence prognosis in this context (Appendix A). In a multivariate model ran in TN-AML subset, that included pre-treatment (age, WBC count, karyotype) and treatment-related (induction regimen, allogeneic transplant) covariates, IPS was the only independent variable associated with OS (HR 3.75; 95% CI 1.5–9.6; *p* = 0.006) and an independent variable for DFS (HR 2.2; 95% CI 0.9–5.5; *p* = 0.046) and EFS (HR 2.50; 95% 1.2–5.1; *p* = 0.012), together with allogeneic transplant (Table 3). In patients achieving CR, an MRD evaluation by MFC was available for 62 patients. MRD negativity rate had a non-significant trend in IPS groups, being 50.0% (*n* = 10/20) and 38.1% (*n* = 16/42) in IPS_0 and IPS_1-2, respectively (*p* = 0.41). 

### 3.5. Interaction with Treatment Covariates in TN-AML

We investigated any potential effect of treatment-dependent covariates (induction containing HDAC versus SDAC; allogeneic HSCT as time-dependent) on outcome in IPS-related groups (Appendix A). As per EFS, IPS_0 patients showed a non-significant trend for a higher impact of HDAC-including induction (HR 0.28; 95% CI 0.0–2.3; *p* = 0.23) in comparison with IPS_1–2 (HR 0.92; 95% CI 0.5–1.6; *p* = 0.77). Both IPS categories had an EFS benefit from allogeneic transplant with HR values of 0.11 (*p* = 0.048) and 0.34 (*p* = 0.054) for IPS_0 and IPS_1-2, respectively.

## 4. Discussion

The clinical management of patients with AML is progressively diversifying the therapeutic approaches because of the development of novel agents and the refinement of prognostic stratification driven by an increasing knowledge of the genetic background [26]. In the upfront treatment, the kinase inhibitor Midostaurin has improved long-term outcomes in patients bearing FLT3 mutations [27] and benefit from the addition of Gentuzumab Ozogamicin to conventional chemotherapy was achieved mostly in favorable-risk karyotype [28,29,30,31]. Still, a non-negligible fraction of patients places in a genetically undefined category. This subset collects by exclusion a variety of cases with heterogeneous biological and clinical characteristics, which lack specific molecular features either for targeted drugs or for proper risk assessment. According to WHO, some of these cases are classified in the AML–MRC subgroup based on morphologically assessed dysplasia, and are therefore stratified as adverse-risk [2]. Such an attribution, driven by a non-standardized technique, is still debated, especially due to the clinical decisions it would imply [5,7,9,13].

In order to address this issue, we retrospectively studied MLD by means of MFC in a large series of AML patients intensively treated at our Center. The correlation with morphology we observed in the unselected analysis reiterated the capability of the MFC-based approach at highlighting dysplasia in AML. As expected, MLD data provided no insight into prognosis when considered overall (Figure 2) or within well-defined genetic categories (Appendix A). Of interest, IPS-related dysplasia conveyed significant prognostic information when we focused on genetically undefined patients (TN-AML), representing about 25% of our series. In this context, the lack of dysplastic features (IPS_0) correlated with significantly higher CR rate and longer survival compared to patients showing dysplasia in one (IPS_1) or both (IPS_2) evaluated cell lineages (Figure 3). The impact of IPS category maintained its validity after censoring at allogeneic HSCT and in a multivariate analysis including baseline and treatment-related covariates. With respect to the morphology, the application of MFC may be particularly useful in this setting, primarily overcoming the operator-dependent variability in interpretation of dysplastic cell features, and providing the opportunity for quantification of the extent of deviation from normal. Based on the sensitivity set for the detectability of a cell compartment (i.e., ≥0.01% of global cells), the vast majority of our patients were evaluable for dysplasia. The score we adopted has been explored in the MDS setting [14] and can be relatively easy to derive from standard MFC approaches that are employed at the time of diagnostic workup.

Our results could fit with the need for further information for improving the clinical management of patients with TN-AML. In the induction phase, we did not observe any differential benefit in IPS-defined subgroups deriving from the adoption of standard versus high dose cytarabine. The novel therapeutic agent CPX-351, providing a liposomal encapsulation of cytarabine and daunorubicin at a fixed 5:1 molar ratio, appeared superior to standard “3+7” regimen in AML-MRC [32,33]. An analogous advantage was not highlighted in other AML categories [32] and was not demonstrated in patients for whom MRC is defined by morphologic dysplasia, who were excluded from the phase 3 trial [33]. In this view, a more reliable, MFC-based appraisal of dysplasia might aid at identifying patients that potentially benefit from the adoption of CPX-351, as well as other emerging treatment combinations (i.e., BCL2 inhibitors plus hypomethylating agents) [34], in the scarcity of genetic predictors of response.

As per allogeneic HSCT, as well as other therapeutic implications, we recognize the limits of a retrospective study covering a long time period, with consequent changes in risk assessment and treatment allocation. The worse prognosis that we observed for dysplastic cases (IPS_1-2) within TN-AML might suggest an intensification of post-CR phase, by offering HSCT to these patients. In our study, the actual delivery of the transplant yielded a favorable effect on relapse prevention, independently of stratification according to IPS (Appendix A). This finding is consistent with the data from some retrospective, uncontrolled experiences [35,36] and by donor versus no-donor comparison [37] in the same category of patients (i.e., TN-AML).

Beyond its retrospective design, some limitations of our study were the lack of a validation cohort and all the molecular data needed to align with the 2017 ELN stratification [26]. Based on the latter, a small percentage of TN-AML cases would have been reassigned for disease risk due to gene mutations involving *TP53*, *ASXL1*, *RUNX1*. Since all these genotypes correlate to adverse prognosis, it could reasonably be assumed that the enucleation of such cases would not alter the overall results, and, in particular, the relatively good outcome of the IPS_0 group.

## 5. Conclusions

In our retrospective study of intensively treated AML cases, we demonstrated a significant prognostic effect exerted by dysplasia as assessed by flow cytometry in a subgroup featuring a lack of genetic determinants (TN-AML). In this context, our data could help address the relative unmet needs in terms of risk assessment and treatment strategy, and provide insight into the prediction of response in the rapidly evolving therapeutic scenario of AML.

## Figures and Tables

**Figure 1 cancers-12-03196-f001:**
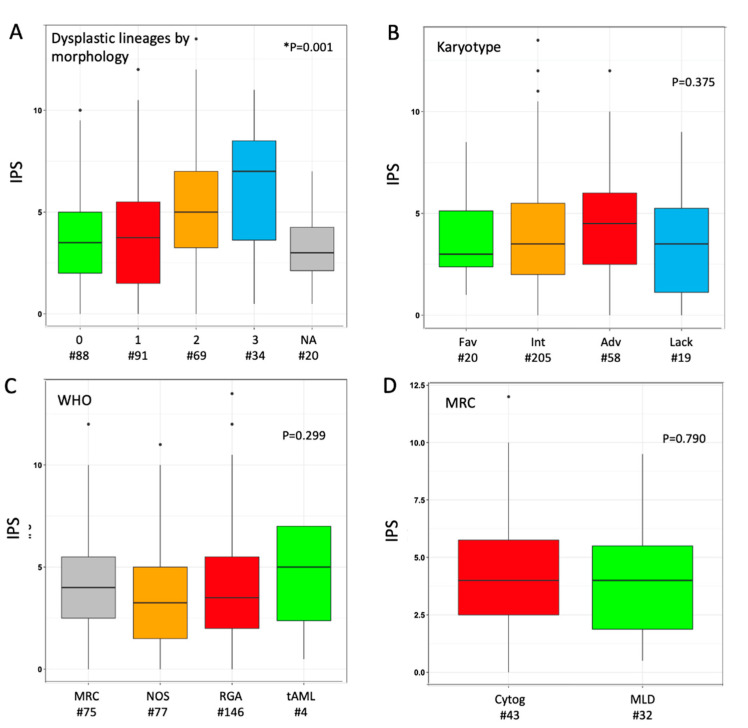
Distribution of immunophenotypic score (IPS) according to disease categorization. Box plot diagrams display the distribution of IPS values according to (**A**) number of dysplastic cell lineages by morphology, (**B**) karyotype risk, (**C**) WHO subset, (**D**) criterion for assignment to AML-MRC (either cytogenetic or morphological). Boxes represent the interquartile range that contains 50% of the cases, the horizontal line in the box marks the median. Dots are outliers. Comparisons were carried out by Mann–Whitney or Kruskal–Wallis or test (depending on number of categories, either two or more). * In panel (**A**), twenty cases were not assessable (NA) for any of the three cell lineages by morphology; in this analysis, Kruskal–Wallis test was carried out excluding NA cases. Abbreviations: IPS, immune-phenotypic score; Fav, favorable; Int, intermediate; Adv, adverse; Lack, lack of growth; WHO, World Health Organization; MRC, myelodysplasia-related changes; NOS, not otherwise specified; RGA, recurrent genetic abnormalities; tAML, therapy-related AML; Cytog, defined by MDS-related cytogenetic abnormality; MLD, defined by multilineage dysplasia at morphology; wt, wild type; MUT, mutated; ITD, internal tandem duplication; DM, double-mutated.

**Figure 2 cancers-12-03196-f002:**
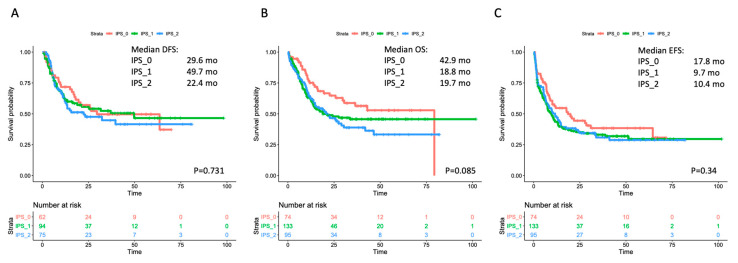
Survival estimates according to IPS in the overall cohort. Major outcomes as assessed by Kaplan-Meier curves for (**A**) disease-free survival (DFS), (**B**) overall survival (OS), and (**C**) event-free survival (EFS) according to number of dysplastic cell lineages by immunophenotypic score (IPS). The curves of patients with no dysplastic lineages (IP_0) are depicted in red; the curves of patients with one dysplastic lineage (IPS_1) are depicted in green, and the curves of patients with both (neutrophil and erythroid) dysplastic lineages (IPS_2) are depicted in blue. Median survival estimates are reported in months. *p* values were calculated through the log-rank test.

**Figure 3 cancers-12-03196-f003:**
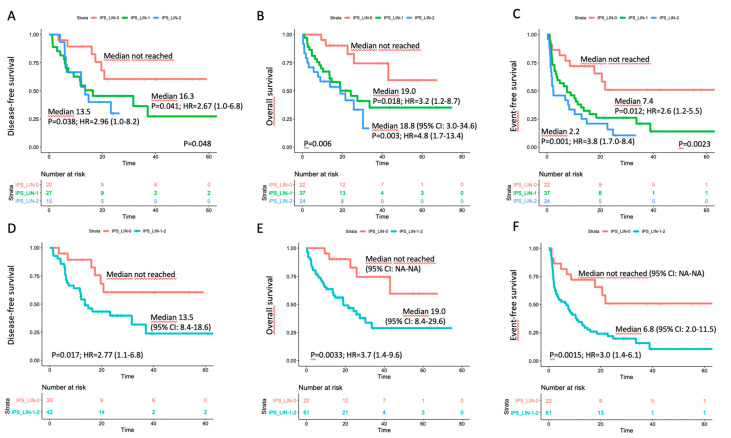
Survival estimates according to IPS in the triple-negative subset. Major outcomes as assessed by Kaplan–Meier curves for (**A**,**D**) disease-free survival, (**B**,**E**) overall survival, and (**C**,**F**) event-free survival (EFS) according to number of dysplastic lineages by immunophenotypic score (IPS) in triple-negative (TN-AML) patients. In panels (**A**–**C**), patients were separated in three categories; in panels D-F, IPS_1 and IPS_2 cases were grouped.

**Figure 4 cancers-12-03196-f004:**
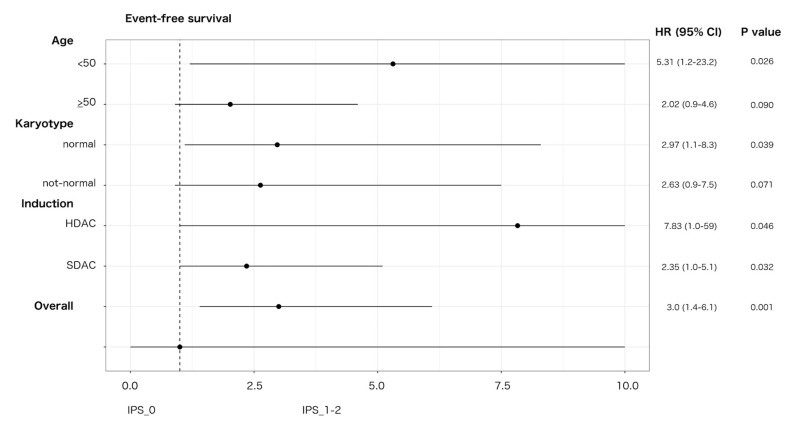
Effect of IPS on EFS in triple-negative AML. Forest plot depicting the effect of IPS-related group (IPS_0 versus IPS_1-2) on EFS in triple-negative AML according to main patient, disease characteristics and induction intensity. HR, hazard ratio; 95% CI, confidence interval; SDAC: standard-dose cytarabine; HDAC: high-dose cytarabine. Upper 95% CI HR values were rounded to 10 for sake of graphical resolution.

**Table 1 cancers-12-03196-t001:** Characteristics of patients in the overall cohort.

	Overall n = 302
**Age, median (range)**	56 (18–75)
**WBC, ×10^9^/L, median (range)**	13.9 (0.6–435.0)
**Hb, g/dL, median (range)**	9.0 (3.4–14.9)
**Plt, ×10^9^/L, median (range)**	50 (3–373)
**Bone marrow blasts, %, median (range)**	90 (20–100)
**Karyotype, n (%)**	
Favorable	20 (6.6)
*t(8;21)/RUNX1-RUNX1T1*	11 (3.6)
*inv(16)/CBF-MYH11*	9 (3.0)
Normal	174 (57.6)
Intermediate, non-normal	31 (10.3)
Adverse	58 (19.2)
*complex*	38 (12.6)
*chromosomes 5/7 abnormalities*	14 (4.6)
*chromosome 3 abnormalities*	4 (1.3)
*t(6;9)*	2 (0.7)
Lack of growth	19 (6.3)
**Molecular genetics, n (%)**	
NPM1-mutated	115 (38.0)
FLT3-ITD	73 (24.2)
FLT3-TKD	7 (2.3)
CEBPA-DM	13 (4.3)
**WHO Classification, n (%)**	
Recurrent genetic abnormalities	146 (48.3)
Therapy-related	4 (1.3)
Myelodisplasia-related changes	75 (24.8)
Not otherwise classified	77 (25.6)
**ELN 2010 risk groups, n (%)**	
Favorable	90 (29.8)
Intermediate-1	129 (42.7)
Intermediate-2	25 (8.3)
Adverse	58 (19.2)

Differences between treatment groups were evaluated using Mann–Whitney test for continuous variables and Fisher exact tests or χ^2^ for categorical variables. Values in bold are statistically significant (*p* < 0.05). Abbreviations: WBC, white blood cells; Hb, hemoglobin, Plt, platelets; CEBPA-DM: double-mutated; ELN, European Leukemia Net.

**Table 2 cancers-12-03196-t002:** Characteristics of patients according to IPS in triple-negative disease subset.

	IPS_0 n = 22 (26.5%)	IPS_1-2 n = 61 (73.5%)	*p* Value
Age, median (range)	56.5 (35–72)	56 (22–73)	0.958
WBC, x10^9^/L, median (range)	2.2 (0.6–36.9)	5.7 (1.0–220)	0.018
Hb, g/dL, median (range)	9.5 (4.8–13.3)	9.0 (4.6–14.9)	0.350
Plt, x10^9^/L, median (range)	72 (16–281)	54 (3–271)	0.070
Bone marrow blasts, %, median (range)	85 (20–100)	80 (20–100)	0.910
Karyotype, n (%)			
Normal	11 (50.0)	46 (75.4)	0.034
Intermediate, non-normal	8 (36.4)	12 (19.7)	0.148
Lack of growth	3 (13.6)	3 (4.9)	0.186

Differences between treatment groups were evaluated using Kruskal-Wallis test for continuous variables and Fisher exact tests or χ^2^ for categorical variables. Values in bold are statistically significant (*p* < 0.05). Abbreviations: IPS, Immuno-Phenotypic Score; WBC, white blood cells; Hb, hemoglobin, Plt, platelets.

**Table 3 cancers-12-03196-t003:** Multivariate analyses (Cox regression) for major outcomes in triple-negative AML.

	MULTIVARIATE ANALYSIS									
	HR (95% CI)	*p*	HR (95% CI)	*p*	HR (95% CI)	*p*	HR (95% CI)	*p*	HR (95% CI)	*p*
	Step 1		Step 2		Step 3		Step 4		Step 5	
Disease-free survival										
Age (</≥50 y)	0.62 (0.25–1.57)	0.31	0.64 (0.26–1.56)	0.33	0.65 (0.27–1.58)	0.34	-	-	-	-
WBC (cont.)	1.00 (1.00–1.00)	0.19	1.00 (1.00–1.00)	0.19	1.00 (1.00–1.00)	0.18	1.00 (1.00–1.00)	0.21	-	-
Karyotype (normal vs. not)	1.42 (0.53–3.78)	0.48	1.32 (0.58–3.03)	0.51	-	-	-	-	-	-
Induction (SDAC vs. HDAC)	0.88 (0.35–2.23)	0.79	-	-	-	-	-	-	-	-
HSCT (T-dep.)	0.18 (0.06–0.52)	0.002	0.18 (0.06–0.52)	0.002	0.18 (0.06–0.53)	0.002	0.22 (0.08–0.59)	0.002	0.24 (0.09–0.63)	**0.046**
IPS (_0 vs. _1–2)	3.04 (1.04–8.90)	0.042	2.89 (1.07–7.80)	0.036	2.53 (1.01–6.23)	0.046	2.43 (0.98–6.00)	0.046	2.21 (0.89–5.45)	**0.003**
**Overall survival**										
Age (</≥50 y)	0.97 (0.45–2.10)	0.95	-	-	-	-	-	-	-	-
WBC (cont.)	1.00 (1.00–1.00)	0.70	1.00 (1.00–1.00)	0.70	1.00 (1.00–1.00)	0.73	-	-	-	-
Karyotype (normal vs. not)	2.07 (0.92–4.63)	0.08	2.06 (0.93–4.60)	0.08	1.93 (0.97–3.84)	0.06	1.94 (0.98–3.86)	0.06	1.76 (0.91–3.42)	0.095
Induction (SDAC vs. HDAC)	0.87 (0.38–2.00)	0.75	0.88 (0.40–1.96)	0.75	-	-	-	-	-	-
HSCT (T-dep.)	0.59 (0.25–1.40)	0.23	0.59 (0.26–1.34)	0.21	0.58 (0.26–1.29)	0.18	0.58 (0.26–1.30)	0.19	-	-
IPS (_0 vs. _1-2)	4.95 (1.73–14.1)	0.003	4.93 (1.74–14.0)	0.003	4.69 (1.74–12.6)	0.002	4.60 (1.72–12.3)	0.002	4.61 (1.74–12.2)	**0.002**
**Event-free survival**										
Age (</≥50 y)	0.61 (.33–1.13)	0.12	0.62 (0.34–1.14)	0.12	0.66 (0.37–1.18)	0.16	0.69 (0.39–1.22)	0.20	-	-
WBC (cont.)	1.00 (1.00–1.00)	0.34	1.00 (1.00–1.00)	0.35	1.00 (1.00–1.00)	0.41	-	-	-	-
Karyotype (normal vs. not)	1.17 (0.60–2.27)	0.65	-	-	-	-	-	-	-	-
Induction (SDAC vs. HDAC)	0.76 (0.39–1.46)	0.41	0.81 (0.45–1.47)	0.48	-	-	-	-	-	-
HSCT (T-dep.)	0.21 (0.07–0.60)	0.003	0.21 (0.08–0.61)	0.004	0.21 (0.07–0.59)	0.003	0.22 (0.08–0.61)	0.004	0.26 (0.09–0.69)	**0.007**
IPS (_0 vs. _1–2)	2.87 (1.31-6.28)	0.009	2.69 (1.29–5.58)	0.008	2.62 (1.27–5.41)	0.009	2.51 (1.21–5.16)	0.013	2.50 (1.22–5.14)	**0.012**

In the final step of the multivariate analysis, statistically significant values (*p* < 0.05) are indicated in bold. Abbreviations: HR, Hazard Ratio; CI, Confidence Interval; WBC, white blood cell; cont., as continuous variable; SDAC, standard dose cytarabine; HDAC, high dose cytarabine; HSCT, allogeneic transplant; T-dep, as time-dependent covariate; IPS, Immuno-Phenotypic Score.

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
