# Peer review of "Multilineage Dysplasia as Assessed by Immunophenotype in Acute Myeloid Leukemia: A Prognostic Tool in a Genetically Undefined Category"

_cancers, 2020, doi:10.3390/cancers12113196_

Round 1

Reviewer 1 Report

In this study Mannelli et al. investigated multi-lineage dysplasia (MLD) by multiparameter flow cytometry (MFC) in 302 AML patients treated with intensive chemotherapy and compared the results with with morphology. MLD data, estimated through an immune-phenotypic score (IPS), did not provide insight into prognosis when considered overall nor within well-defined genetic categories. However, IPS- related dysplasia conveyed significant prognostic information in genetically undefined patients, triple-negative for NPM1, FLT3 and CEBPA (TN-AML). The lack of dysplastic features (IPS_0) correlated with higher complete response rate and longer survival compared to patients showing dysplasia in one or both (neutrophil and erythroid) cell lineages. The impact of IPS category maintained its validity after censoring at allogeneic HSCT and in a multivariate analysis including baseline and treatment-related covariates. These results may suggest the value of MFC in risk assessment and treatment strategy of AML, and prediction of response in the subgroup of AML.

I have found the following week points of the paper.

  1. The patients included into the study were observed from April 2004 to April 2017 when the diagnostic and therapeutic approaches evolved significantly.
  2. The retrospective nature of this study is it major limitation
  3. Other limitation is the lack of a validation cohort
  4. Karyotype analysis is lacking
  5. Several, prognostically important genes were not included (PML-RARA, RUNX1-RUNX1T1, CBFB-MYH11, TP53, RUNX1, ASXL1 IDH1 and IDH2).
  6. Taking into account that several genetic parameters were not included in this study, the prognostic value of MFC seems to be low

Reviewer 2 Report

Multilineage dysplasia as assessed by immune-phenotype in acute myeloid leukemia: a prognostic tool in a genetically undefined category

Francesco Mannelli1,2, Sara Bencini1,3, Matteo Piccini1, Giacomo Gianfaldoni1, Maria Ida 4 Bonetti1,3, Benedetta Peruzzi3, Roberto Caporale3, Barbara Scappini1, Fabiana Pancani1, Vanessa 5 Ponziani1, Leonardo Signori1, Michela Zizza1, Francesco Annunziato3, Alberto Bosi1 6

1SOD Ematologia, Università di Firenze, AOU Careggi, Firenze, Italy; 2Centro Ricerca e Innovazione Malattie 7 Mieloproliferative (CRIMM), AOU Careggi, Firenze, Italy; 3Centro Diagnostico di Citofluorimetria e 8 Immunoterapia, AOU Careggi, Firenze, Italy 9

* Correspondence: Francesco Mannelli, MD, SOD Ematologia, Centro Ricerca e Innovazione Malattie 10 Mieloproliferative (CRIMM), AOU Careggi, Università di Firenze, Largo Brambilla 3, 50134 Firenze, Italy; 11 Phone/fax: +39 055 7947824; e-mail: francesco.mannelli@unifi.it

REVIEW COMMENT:

The study aims to assess trilineage dysplasia by multiparameter flow cytometry (MFC) in a large cohort of 302 AML patients and to correlate with other clinicopathologic parameters and outcomes. The results indicated that immunophenotypic score (IPS)-related dysplasia provided valuable prognostic information, in particular in AML patients genetically undefined or AML triple-negative for NPM1, FLT3, and CEBPA. The absence of dysplastic features (immuno-phenotypic score, IPS_0) showed a higher CR rate and a longer survival time when compared to patients with single or bi-lineage (neutrophil and erythroid) dysplasia. The manuscript is well written and easy to follow. The provided supplemental information helps understanding the study. It is recommended to publish in the peer review journal after revision.

Major critiques:

  1. According to WHO classification, to diagnose AML with myelodysplastic related changes9AML-MRC) must meet any one of the following three criteria: 1) preexisting history of MDS (myelodysplastic syndromes), 2). harboring MDS-related cytogenetic abnormalities and 3) morphologic dysplasia identified in >=2 lineage cells, 50% for each line. To correlate lineage-specific dysplasia in AML beyond AML-MRC by using flow cytometry and correlate with clinical parameters is under-investigated as other parameters might override dysplasia e.g., FLT-3 mutation, RUNX1 mutation, TP53 mutation, etc. For the study, to subclassify the AML at first based on WHO is necessary to avoid a selection bias. Given different outcome and treatment strategies, ideally, we should analyze IPS under each AML subcategory as listed including AML-MRC, AML with recurrent cytogenetic or molecular abnormalities, and AML, NOS (refer to the lines 130-160). The role of dysplasia is assumed to be different according to the AML type. Therapy-related AML itself has better excluded given a limited case number.
  2. Maybe missed, please add the detailed about “immunophenotypic score” 
  3. ELN 2010 AML risk stratification has been updated. It is recommended to replace it with ELN 2017 in the study and correlate with flow cytometric data if data is available, 
  4. Dysplasia is just a finding that can be assessed by morphology or flow cytometry. But genetic/molecular background plays a more critical role in disease prognosis. Other NGS data, in addition to FLT3, CEBPA, and NMP1 gene mutation, should be correlated if data is available. Also, in the study, it concluded that IPS-0 (no dysplasia) had a better clinical outcome than those without dysplasia (abstract), which is inaccurate and a bit misleading. We have better know the distribution of the patients with IPS-0 in each AML subtype, e.g., IPS-0 is more frequently identified in AML-NOS than AML-MRC. By all means, AM-MRC itself shows a worse clinical outcome

Reviewer 3 Report

In this manuscript dr Mannelli et al. developed an immune-phenotypic score (IPS) based on dysplasia in AML negative for FLT3 NPM1 and CEPBA mutations.

They found that the IPS does not provide further information on the prognosis if applied to all cases of LAM, by contrast it is significant if applied to  particular subgroups such as triple negative. In this patients the IPS score O correlates with higher CR rate and better OS. 

The study is scientifically sound , the message is new and  lead to a step forward in the management of patients with AML. The study is well conducted, the series of patients is adequate.

The limitations of the study were clearly indicated by the authors in the discussion section. 

Round 2

Reviewer 1 Report

I have not morecomments